# From Keratoma to Anaplastic Malignant Melanoma in a Horse’s Hoof

**DOI:** 10.3390/ani12223090

**Published:** 2022-11-09

**Authors:** Elżbieta Stefanik, Kamil Górski, Bernard Turek, Olga Drewnowska-Szczepakowska, Katarzyna Kliczkowska-Klarowicz, Aleksandra Stefanik

**Affiliations:** 1Department of Large Animal Diseases and Clinic, Institute of Veterinary Medicine, Warsaw University of Life Sciences, 02-787 Warsaw, Poland; 2Division of Pathology, Department of Pathology and Veterinary Diagnostis, Institute of Veterinary Medicine, Warsaw University of Life Sciences, Nowoursynowska 159c, 02-776 Warsaw, Poland; 3Zagłębie Oncology Centre, Szpitalna 13, 41-300 Dąbrowa Górnicza, Poland

**Keywords:** anaplastic malignant melanoma, equine oncology, oncology

## Abstract

**Simple Summary:**

Melanocytic origin neoplasms are most common in older and gray horses, but their nature may differ from those of non-gray horses. In this case report, we describe the repeated excision of a hoof lesion, which was initially diagnosed as a keratoma with melanocytic grains, and evolved over time into an anaplastic malignant melanoma.

**Abstract:**

Melanomas in horses are most often associated with gray, older horses with an average age of over 16 years. Anaplastic malignant melanoma, however, can very rarely affect non-gray horses. Herein, we report a case of a 16-year-old Wielkopolski gelding with a chronic lameness caused by a mass in the hoof. The first resection of the lesion and histopathological examination confirmed the presence of a keratoma. The regrown mass and persistent lameness resulted in another mass resection. The second histopathological examination result suggested a neoplastic growth of melanocytic origin with a low histological malignancy. Less than 2 years after the first resection, the horse returned to the clinic with deformation of the hoof capsule and severe lameness. The result of the third histopathological examination indicated low-differentiated malignant neoplasm. The result of the immunohistochemically test indicates a tumor of melanocytic origin with high malignancy.

## 1. Introduction

Neoplasms are relatively rare in horses and amount to about 3% of the problems reported in veterinary practice [1]. It is estimated that approximately half of them are neoplasms of skin origin, including sarcoids (54.4%), followed by squamous cell carcinoma (18.3%). Melanocytic tumors, such as gray horse melanoma, are in third place (5.4%), melanocytoma are only 4.1%, and malignant melanoma is 0.7% of the total neoplasms. In clinical practice, melanomas are most often found in gray horses, but their characteristics significantly differ from those in horses of other color variations. In non-gray horses, we can distinguish benign and malignant variants, with a higher probability of developing a malignant phenotype [2]. Recent clinical reports indicate that all melanocytic tumors in horses should be treated as malignant neoplasms [3,4]. The malignancy of the tumor is determined by its aggressive and invasive behavior. In the case of melanocytic neoplasms in horses, not only histological malignancy should be taken into account, but also the ability to spread locally (melanomatosis–found mainly in gray horses) [4]. There are also neoplasms in equids which, due to their localization, should be treated as malignant, even without histological malignancy, as in the described case of melanoma in the internal laminae of the hoof of a mule [3]. Due to the function of the horse’s hoof, the appearance of any space-occupying mass within the hoof capsule may cause a clinically significant problem. Such situations are rare and most often involve the appearance of a keratin soft mass called keratoma [5]. Due to the fact that most clinical symptoms are related to the presence of the mass itself and not to the nature or origin of the lesion, in cases of differential diagnoses, rare neoplastic lesions should be considered.

## 2. Materials and Methods

### Case Presentation

A 16-year old Wielkopolski (Polish warmblood), chestnut gelding, weighing 450 kg, was referred to the Equine Clinic of the Warsaw University of Life Sciences for an orthopedic consultation due to lameness, which had been gradually increasing over the prior two months. Observation of the horse in motion revealed supporting limb lameness of the right front limb, rated at 4 points on a 5-point scale (according to the AAEP Grading System). A close visual examination showed the presence of a purulent fistula in the area of the coronary band and deformation of the anterior wall of the hoof capsule (Figure 1). Palpation also revealed increased pulsation of the digital arteries and increased warmth of the hoof. During the hoof tester examination, a clearly marked withdrawal reflex was found during the application of pressure in the area of the front part of the hoof wall. After superficial cleaning and trimming of the sole, white line distortion and deviation towards the frog apex were found (Figure 2). Perineural anesthesia was performed to confirm that the cause of the lameness was localized in the foot. The lameness decreased visibly after the basilar sesamoid nerve block (2 mL of lignocaine HCl 2%) was performed. Radiographic examination of the affected foot in upright pedal technique revealed a circular well-demarcated area of the radiolucency along the solar margin of the third phalanx (Figure 3). For comparison, the radiograph of the opposite limb was also performed in the same projection. Based on the identified changes, the keratoma was diagnosed.

Partial hoof resection was chosen as the best surgical approach. In order to soften the hoof capsule and ensure aseptic conditions, a wet dressing with 1% povidone iodine was applied the day before the procedure. The procedure was performed under general anesthesia. Low 4-point block was performed with bupivacaine HCl 0.5% (2 mL over each palmar metacarpal nerve and 5 mL over each palmar nerve). The keratoma was exposed by removing the hoof wall, for which a cast saw was used by making 2 parallel cuts on both sides of the expected keratoma. The third cut line ran above the defect, and the fourth cut was in sole on the border of the white line. A section of the hoof wall was separated and lifted using forceps. The keratoma with a margin of healthy tissue was removed with a laurel knife (Figure 4). Affected soft tissues and necrotic foci on the area of the coffin bone were curetted. The size of the defect was estimated at 3 × 5 cm. Exposed laminae corium were covered with iodoform, and sterile rollers made of gauze pads were applied. The rollers were packed tightly, the first layer in parallel from the long axis of the limb and the other perpendicularly to layer one. The dressing was covered with medical gauze, then wrapped with a thick layer of lignin, over which an elastic band was applied.

The fragment collected during the operation was sent for histopathological examination. In the obtained results, the diagnosis indicated keratoma. The collected sample also showed the presence of numerous melanocytes, filled with melanin grains, showing no signs of cellular atypia or signs of proliferation (Figure 5). A pressure bandage was used throughout the treatment, analogous to that applied immediately after surgery, until the granulation tissue was completely covered with the new horn. The horse was shod in a full-bar horseshoe to stabilize the hoof. In the initial phase of treatment, the horse remained in the stall, and after 6 weeks, controlled walk-in-hand walks were introduced. During this time, the lameness remained at the level of 3/4 degree. Seventy-five days after surgery, a control X-ray examination was performed, which showed significant coffin bone loss. Later, 116 days after surgery, the horn of the anterior wall had grown to half its height, but its quality was poor and its color was black compared to the rest of the hoof capsule, and some small deformation in the central line of the coronary band was present (Figure 6). After 173 days in the clinic, when the lameness was reduced to a second degree, the decision was made to send the horse home. Further hoof bandage changes, controlled walks in hand, and the use of a special shoe to protect the hoof were recommended.

The horse returned to the clinic with grade 3 lameness 341 days after surgery. Upon arrival, a fistula opening was found near to the edge of the anterior sole, as well as the midline of coronary band. Above the coronary band a clear, painful deformation was also found. A fragment of the anterior wall was resected down to the corium. From the clinical picture it could be concluded that the coffin bone was atrophied. Exposed tissues, especially around the coronary band were pigmented. A fragment was taken for histopathological examination a second time. Within the examined tissues, especially in the wall’s horn tubules, large clusters of spindle-shaped cells filled with melanin grains obscuring the cell nuclei were present. These cells showed a low degree of anisocytosis and anisokaryosis, and mitotic figures were not found. The histopathological examination result suggested a neoplastic growth of melanocytic origin with a low histological malignancy. The therapy and the bandages used were analogous to those described after the first treatment. The horse returned home after 5 days.

The horse was referred to the clinic for computed tomography 510 days after the first surgery and 169 days after the second one, due to the lack of effectiveness in the treatment thus far. Due to the presence of deformations within the coronary band (Figure 7 and Figure 8), a decision was made to resection the hoof wall within the deformation, including the coronary band and the skin fragment above it. A fragment was taken for histopathological examination again. Histopathological examination revealed numerous clusters of neoplastic cells recognized in the dermis, also contiguously to the epidermis. Cells had round and oval nuclei with prominent nucleoli, showed significant anisocytosis and anisokaryosis. There were also numerous mitotic figures (MC = 53 mitoses per 10 HPF; 2.37 mm^2^). Several dark brown melanin granules were visible in the cytoplasm of the majority of neoplastic cells (Figure 9). The histopathological picture indicated a low-differentiated malignant neoplasm–possibly both a squamous cell carcinoma and a high-malignancy melanocytic neoplasm (melanoma). In order to determine the origin of the neoplastic cells, an immunohistochemical examination was performed using antibodies against cytokeratin (Anti-Pan Keratin—AE1, AE3, PCK26—mouse monoclonal antibodies raised against human epidermal keratins), vimentin (mouse monoclonal antibody that confirms anti-Vimentin V9 primary antibodies) and Melan-A (mouse monoclonal primary antibody that confirms anti-MART-1/melan A–A103). The result of the immunohistochemical test indicated a tumor of melanocytic origin with high malignancy (the tumor cells showed a high expression of the Melan-A and anti-vimentin antibodies; result for the cytokeratin was negative) (Figure 10). Bandage changes and medication therapy were the same as after the first lesion removal. After 23 days, the horse was sent home. For 10 months after the deformity resection, the lameness remained at the level of 3/5 and gradually progressed. The owners decided not to euthanize the horse, and the horse was under the constant care of a veterinarian. Palliative treatment consisted of the regular trimming of the hoof and the use of a protective shoe (Figure 11).

Due to severe lameness (5/5 AAEP) and the suffering of the animal, the owners decided to euthanize the horse 880 days after the first treatment. No metastases were found during the post-mortem examination. Post-mortem radiographic examination (Figure 12) and computed tomography (Figure 13) were also performed.

## 3. Discussion

The appearance of any abnormal mass of cells in the hoof, causing pressure on the coffin bone, is manifested by severe lameness (AAEP grade 3–4) regardless of the type of cells [6]. The most common lesion of this type is keratoma [5,7]. Keratoma, considered a benign neoplastic process, results from abnormal proliferation of tissues containing keratin and squamous epithelial cells [5,7,8]. This lesion is located between the hoof wall and the distal phalanx, and the resulting keratin mass may be derived from either the coronary or solar corium [7]. The cause of keratoma is not known, although it is believed that trauma or irritation to the corium may contribute to its creation [5]. The clinical symptoms accompanying the appearance of the lesion are hoof capsule deformation and lameness. The degree of lameness depends on the rate of growth of the pathological horny mass inside the hoof, which gradually leads to increasing pressure on the distal phalanx and the formation of bone resorption areas. Depending on the location of the keratoma, deformation of the white line and its displacement may also occur [9].

Keratomas should be differentiated from even less common neoplasms. Due to the localization of the changes, the clinical symptoms are the same and result from the pressure of the mass on the distal phalanx. So far, squamous cell carcinomas [6,10,11], melanomas [1,5,12,13], malignant tumor of the glomus smooth cell [14], mast cell tumor [15], and vascular hamartoma [6] have been described. In all these cases, due to the clinical symptoms accompanying lesions, keratoma was the first to be incorrectly assumed as the diagnosis, the latest histopathological examinations confirmed the different nature of the lesions.

In our case, a histopathological examination of the primary lesion was performed and confirmed the presence of a keratoma. Numerous melanocytic cells were also present in the histological specimen, but these showed no signs of cellular atypia and proliferation. Due to the typical appearance of the keratoma, the fact that melanocytes may be present in the keratoma, no malignant features were found, and the rarity of melanocytic neoplasms in the horse’s hoof, the diagnosis of keratoma was made. Taking into account the later microscopic appearance of this lesion, it cannot be ruled out that from the beginning it was a melanocytic neoplastic hyperplasia without any signs of histological malignancy, which progressed to melanoma. Although at the time of the initial diagnosis, there was no basis for the diagnosis of a neoplasia of melanocytic origin, based on this case, in keratoma diagnosis with numerous melanocytes, the possibility of melanocytic neoplastic hyperplasia should be considered, despite the lack of malignancy.

In the previously described cases of other neoplastic growths, which in the initial stage were considered to be keratomas, one of the common features was lameness that persisted for a long time after removal of the mass [6]. The described case also confirms this observation–after removal of the lesion, clinical improvement was noted, and the degree of lameness decreased but was not resolved. During the second visit of the horse to the clinic, black pigmentation of the horn was noted during the repeated excision of the lesion. The result of the second histological examination suggested a neoplastic growth of melanocytic origin with a low histological malignancy. A similar situation occurred in the case described by Osborne [6]—black pigmentation of the lesion appeared more marked after removal of the presumptively diagnosed keratoma, prompting the authors to take a biopsy sample. Therefore, in this case it is difficult to say whether the original mass was of melanocytic origin due to the lack of a histopathological examination result and the lack of black pigmentation of the lesion in the beginning. However, it is known, that a melanotic or poorly pigmented tumor may occur in both gray and non-gray horses [2]. In our case, it is difficult to state unequivocally whether the surgical stimulation (the process of inflammation and healing after the first treatment) could be a factor contributing to the neoplastic transformation.

Melanomas are neoplasms that are uncommon in non-gray horses. They occur occasionally in the horse’s foot. However, it is known from the available data that the melanomas that appear in the foot of equines are invasive. Although some authors use the classification of benign and malignant for melanocytic neoplasms in horses [16], as early as 1995 Valentine [1] noted that this division is not sufficient due to the nature of the changes in horses. Based on her research, she divided lesions arising from melanocytic cells in horses into 4 forms: melanocytic nevus, dermal melanoma, dermal melanomatosis, and anaplastic malignant melanoma. These forms are characterized by clinical, histological, and behavioral differences, with the exception of dermal melanoma and melanomatosis, which can only be distinguished on the basis of their clinical features. Of the 57 melanocytic lesions from 53 horses, only two of them had distinctive histopathological features, including a high degree of cellular pleomorphism, variable pigmentation, numerous mitoses, and widespread single-cell invasion of the epidermis. Those anaplastic tumors were found in elderly and non-gray horses, and were associated with metastasis within one year of diagnosis. All cases of these types of melanocytic neoplasms reported so far involved older, non-gray horses and were described in the area of the tail [17], hoof [12,13], and sinonasal region [16].

An analogous type of neoplasm found in humans is nail apparatus melanoma (NAM). It is a rare subtype of melanoma, more commonly found in Asians [18] and Africans (up to 25%) [19], as well as Caucasians (2.8%) [20]. Due to this distribution, it is believed that there are factors that may increase the risk of its occurrence. In humans, the lesions form longitudinal bands or total melanonychia with no nail dystrophy, although in the case of invasive lesions, the nail may be split longitudinally [21]. In humans, as in the equine case described by us, the tumor originates from the abnormal nail matrix melanocytes [21]. In nail apparatus melanomas, the role of standard melanoma prognostic factors is under discussion due to conflicting results. In one of the studies, 64.5% of the patients had positive personal history for acute or chronic trauma of the nail. Another study shows 9.7% of participants revealed an acute trauma that had preceded NAM diagnosis [22]. Although the trauma could not be associated with a pathogenetic event, in described cases it was an incidence that helped in faster diagnosis. However, in many cancers smoldering inflammation increases the risk of cancer [23]. Bormann et al. [24] found trauma to be the most decisive negative prognostic factor in their group of patients. In his study, 75% of participants’ NAM were correlated with traumas. The relationship between UV exposure and NAM is less clear than for superficial spreading melanoma. According to Stern [25], the nail plate completely filtrates UV-B light, and only a minimal amount of UV-A light penetrates the nails. In human medicine, wide local excision or amputation is used in treatment [21].

Although the conservative surgery provides a good option in human [21] and removal of the melanocytoma in horses is considered therapeutic, in the described case and in other similar cases [3], it did not bring the expected, curative result. The difficulty in the excision of a lesion from the hoof is due its anatomical complexity and its spreading between the hoof capsule and the coffin bone. In this location, it is also difficult to effectively apply chemotherapeutic agents, such as cisplatin [3]. Additionally, cisplatin can impair not only tumor growth, but also horn growth. Oral use of cimetidine has also been described for the treatment of malignant anaplastic melanomas, but with no satisfactory results [16]. Tumor size-reducing effect has been found after the use of toremifene (applied transdermally in a topical gel) in the treatment of malignant melanomas in horses, but there are no reports of its use for anaplastic malignant melanoma therapy [4]. Other methods of treating melanomas in horses, such as autochthonous vaccine or interleukin therapy [2,4], have also not been documented for this type of melanoma.

## 4. Conclusions

Due to the fact that the clinical symptoms related to the presence of any mass within the horse’s hoof capsule are the same, regardless of the origin and nature of this mass, one should always be aware of neoplastic growths other than keratomas. The rare case described by us confirms the current thinking that any melanocytic-origin tumors should be treated as malignant, especially when they occur in non-gray horses. Additionally, there are still no effective treatments for such lesions in horses.

## Figures and Tables

**Figure 1 animals-12-03090-f001:**
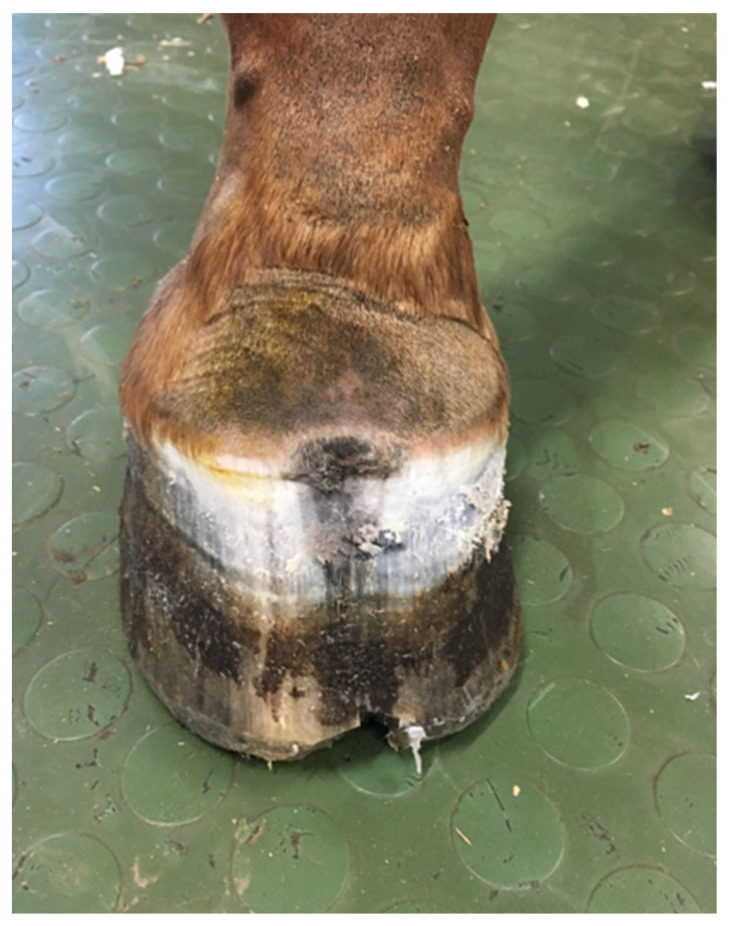
The hoof after superficial cleaning and trimming of the sole, showing visible fistula above the coronary band, deformation, and discoloration of the hoof capsule above at the site of keratoma.

**Figure 2 animals-12-03090-f002:**
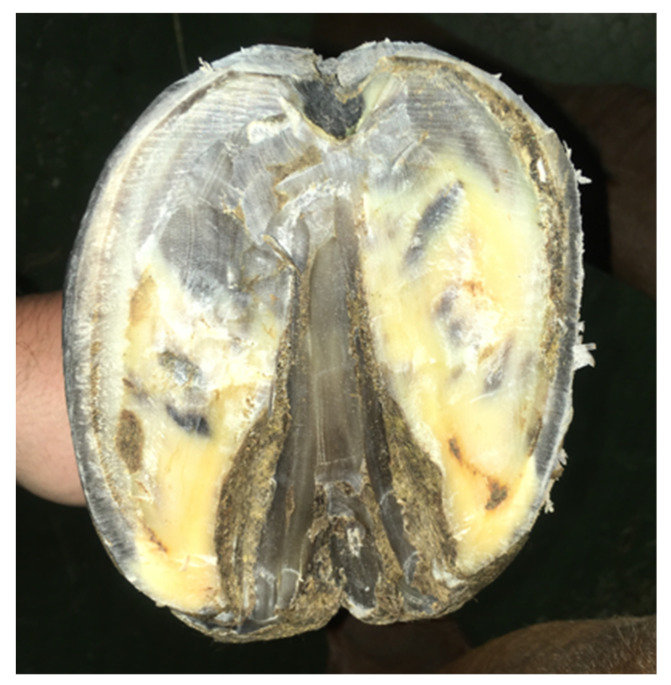
View of the superficially cleaned and trimmed sole of the foot, showing visible separation of the white line.

**Figure 3 animals-12-03090-f003:**
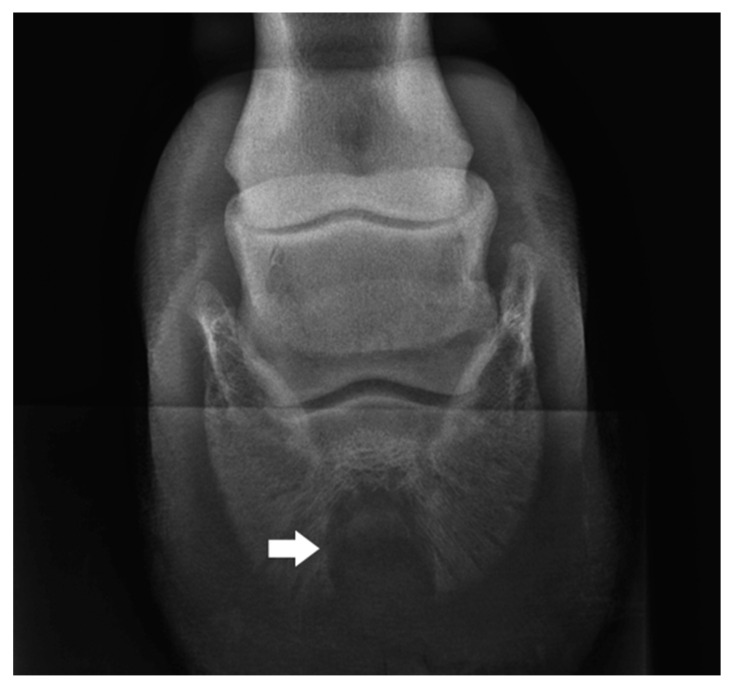
Dorsoproximal-palmarodistal oblique projection with the toe pointing downwards on an Oxspring block, revealing well-defined loss of coffin bone density caused by pressure necrosis from the mass (arrow).

**Figure 4 animals-12-03090-f004:**
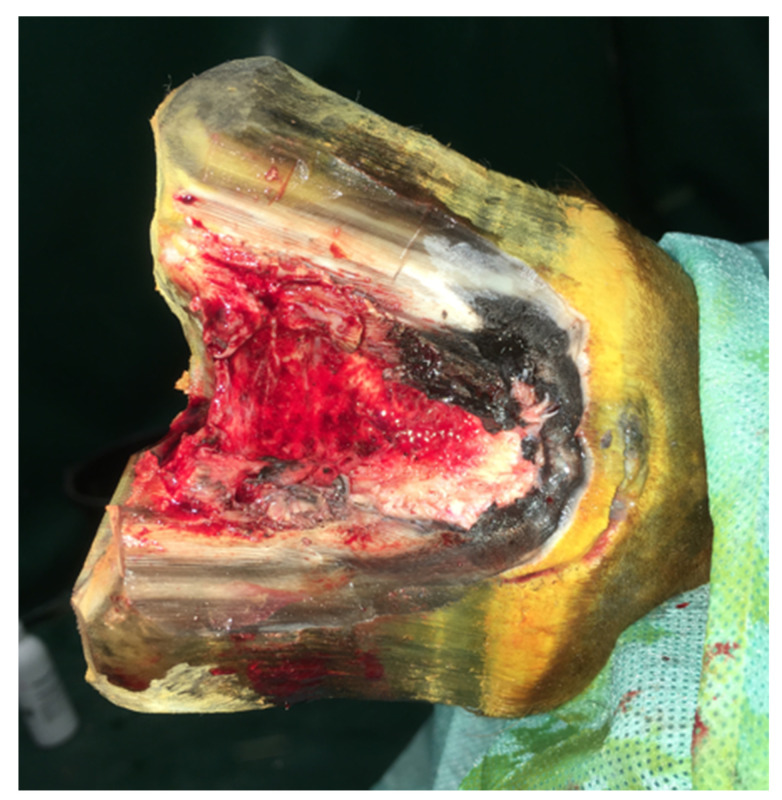
View of the hoof after resection of the keratoma.

**Figure 5 animals-12-03090-f005:**
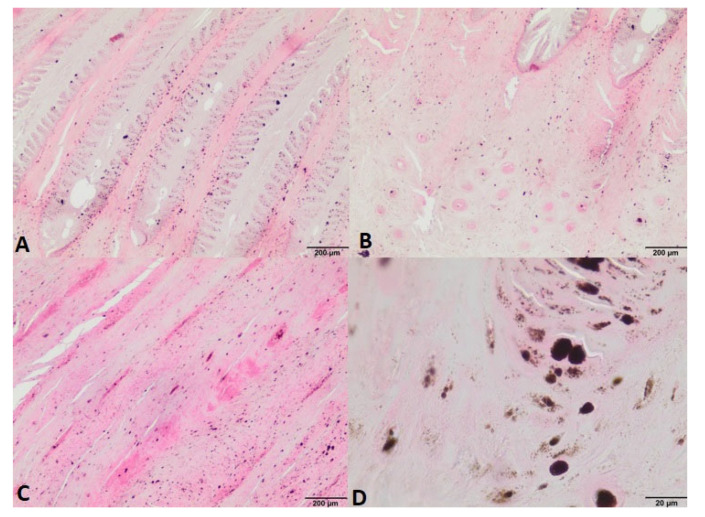
Photomicrograph of the hoof from the first histological examination with a diagnosis of keratoma, showing visible melanocytes, filled with melanin grains, showing no signs of cellular atypia or signs of proliferation. (**A**)–(**C**): HE staining, magn. 40×; (**D**): HE staining, magn. 400×.

**Figure 6 animals-12-03090-f006:**
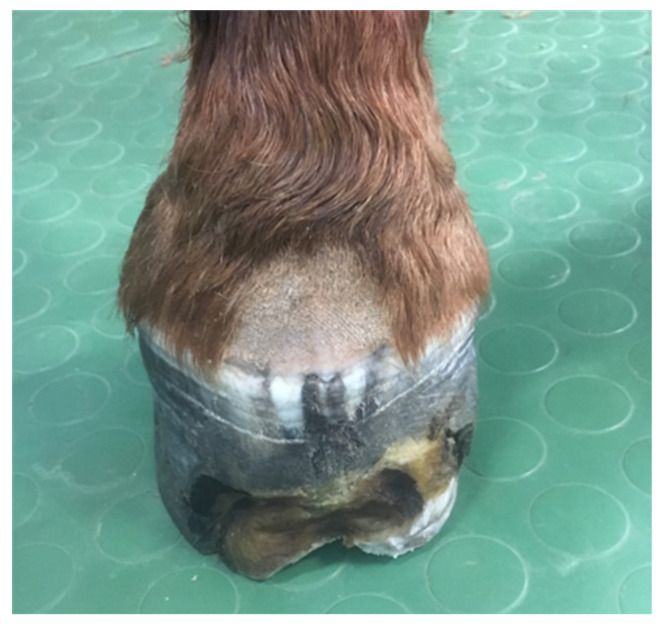
Hoof 116 days after surgery.

**Figure 7 animals-12-03090-f007:**
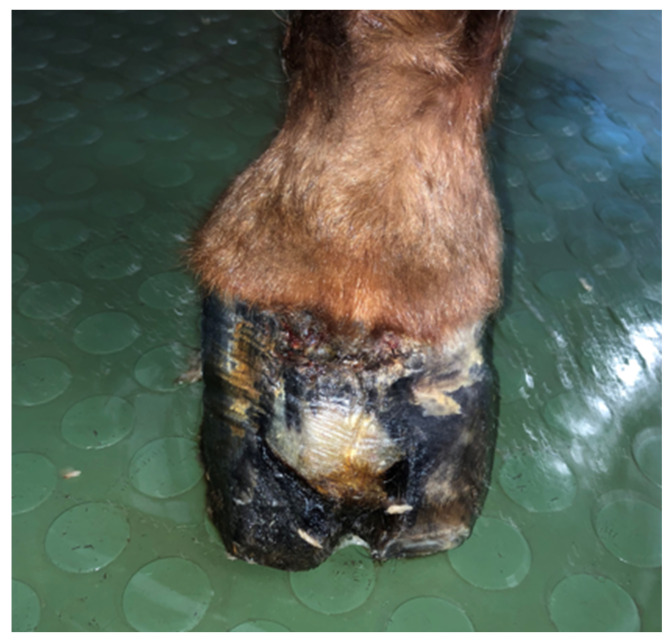
Hoof appearance 510 days after the first treatment and 169 days after the second treatment, showing a visible deformation within the coronary band and hoof capsule.

**Figure 8 animals-12-03090-f008:**
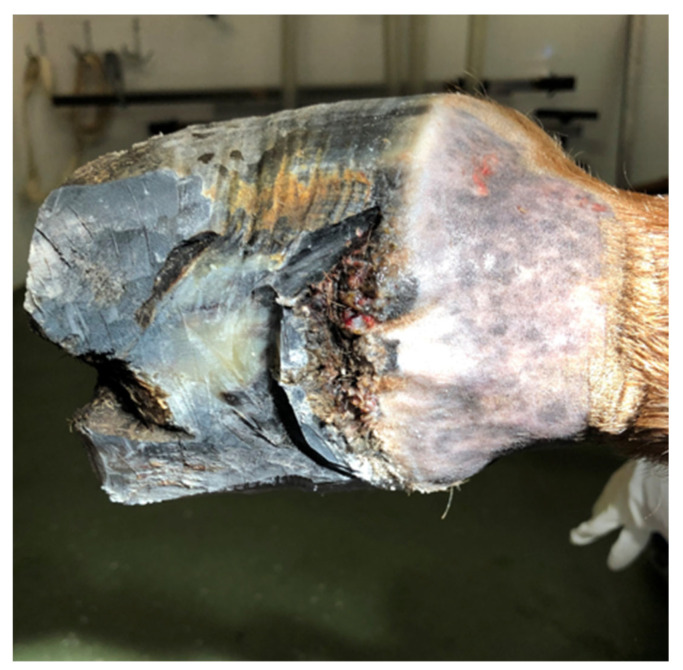
The appearance of the hoof during preparation for the third resection of the lesion, showing a visible dark, rough deformation within the coronary band.

**Figure 9 animals-12-03090-f009:**
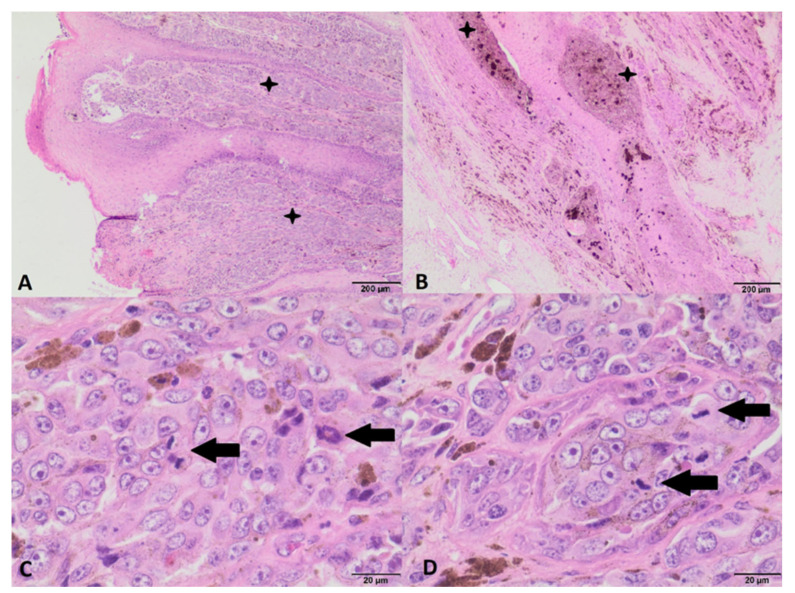
(**A**,**B**): infiltration of neoplastic cells to the dermis (asterisks); HE staining, magn. 40×. (**C**,**D**): neoplastic cells with prominent nucleoli, marked anisocytosis and anisokaryosis, and numerous mitotic figures (arrows); dark brown melanin granules are present in the cytoplasm of many neoplastic cells; HE staining, magn. 400×.

**Figure 10 animals-12-03090-f010:**
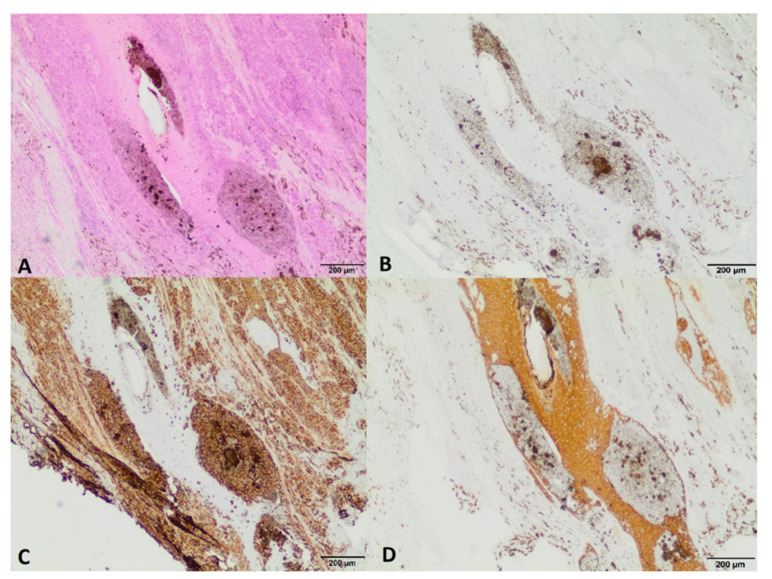
(**A**): infiltration of neoplastic cells to the dermis, HE staining, magn. 40×; (**B**): expression of Melan-A in the cytoplasm of neoplastic cells, magn. 40×; C: expression of vimentin in the cytoplasm of neoplastic cells, magn. 40×; D: negative staining for cytokeratin, magn. 40×.

**Figure 11 animals-12-03090-f011:**
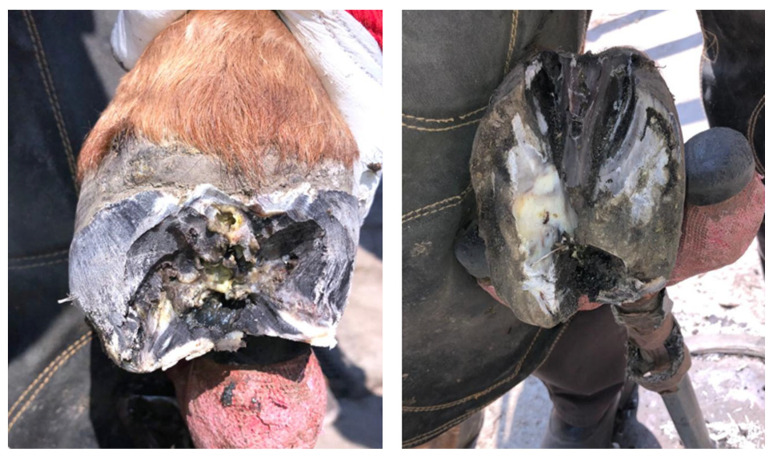
Hoof appearance during trimming 10 months after the third resection of the lesion. A black, irregular deformation is visible under the removed horn. The lysis of the coffin bone was also found.

**Figure 12 animals-12-03090-f012:**
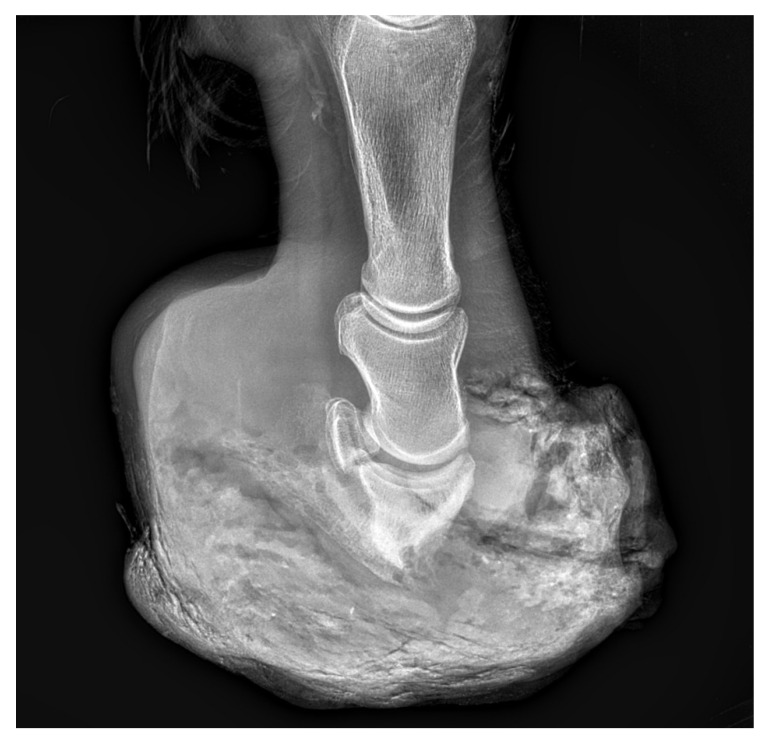
LM projection of the hoof from postmortem examination, revealing a visible large mass in the dorsal part of the hoof capsule pressing against the hoof bone and leading to its lysis.

**Figure 13 animals-12-03090-f013:**
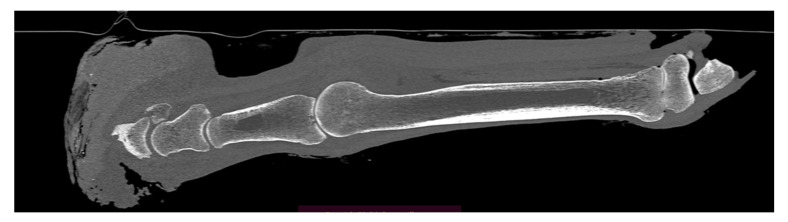
Sagittal CT image (bone algorithm, WW1500, WL300) from postmortem examination, presenting significant deformation and atrophy of the coffin bone, as well as distortion of the hoof capsule with a loss of continuity. Additionally, osseous-cyst like lesion is visible in the long pastern bone.

## Data Availability

Not applicable here.

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
