# Peer review of "From Keratoma to Anaplastic Malignant Melanoma in a Horse’s Hoof"

_animals, 2022, doi:10.3390/ani12223090_

Round 1

Reviewer 1 Report

This paper describes a case report of a horse presented with a neoplasia that was initially diagnosed with a keratoma and then was re-presented with an initial melanocytic lesion that probably evolved in a malignant melanoma. As the authors stated it was not possible to demonstrate that there wasn't a neoplastic melanocytic proliferation at the first surgery and this is a weakness of the case report. 

The melanocytic lesion is well described and it is compatible with a malignant melanoma, but there isn't a description of the antibodies used for immunohistochemistry and if these are validated for the equine species.

The last statement (line 298-299) appears dared, based on just one case and would need a deeper investigation

Author Response

Thank you for your thorough analysis and comments.

Description of the antibodies used for immunohistochemistry has been added.

The last sentence has been changed.

Reviewer 2 Report

This is a well written (although lengthy) case report of a rare, malignant, melanocytic tumor originating from the laminar corium on the foreleg in a non-grey horse. The debilitating and recurring lesion was surgically excised 3 times over a long-term observation period; supposedly, malignant transformation from a pigmented keratoma to a malignant melanoma occurred. The report is completed by post-mortem documentation.  Based on the relative rarity of the lesion, the photographic documentation of a supposed neoplastic transformation, including post-mortem examination, publication as clinical case report is recommended, given satisfactory responses and a major revision.

This reviewer suggests addressing the following prior to publication:

Line 43: “problematic” ? phrase differently in the sense that every space occupying lesion within the hoof is clinically challenging…

Line 46-47: avoid terms like “extremely” (also later on). “forgotten…” rephrase in the sense of “should be considered” …..

Please use only one term for the third phalanx to avoid confusion for the non-specialist reader; although coffin bone and pedal bone are correct terms, use only one (coffin bone).

Material&Methods this heading is usually not used in a case report; please check with editor.

since readers outside Poland do not know what a Wielkopolski Horse is, please add description of colour – also, whether or not affected leg had white spots or skin and hair were pigmented.

Suggest focusing on the lesion from the clinical and pathological standpoint (as you did) and less from the technical-chirurgical-anaesthetic side; (all equine surgeons know how to treat a hoof lesion…): suggest shortening substantially description of anaesthesia, surgical technique (e.g. Esmarch…), post-op; just report what had importance for outcome.

Lines 87-90: The authors state that the histologic diagnosis of a keratoma was made on the first specimen. However, they do not provide an adequate description, saying only that pigmented cells were present, nor do they provide histologic or immunohistochemical images to confirm the diagnosis of keratoma. The authors should include histologic and IHC images showing cytokeratin, vimentin, and Melan A. If not, any link between the excised lesions remains speculative.

Line 109. The numbering of Figures is out of sequence: the first figure on line 109 is numbered Figure 3….followed by Figure 2 and another Figure 3 (with different content than the other Fig. 3. Figure 5 can be omitted, it does not add any useful information; idem Fig. 9.

Fig. 12 and 13: the lesion in the third phalanx is described as “atrophy”; however, given the malignancy of the adjacent lesion, is this not rather a bone lysis than an atrophy ? This should be addressed in the Discussion; either term should only be used if you had histologic confirmation.

Line 147: The authors refer to a "low-differentiated melanoma" and in the Discussion line 224 to a "low-histologic malignancy." The meanings are different, and in the first case indicate a low degree of differentiation, suggesting a neoplasm with a high degree of malignancy, while in the second instance they mean a melanoma with a low degree of malignancy.

Discussion: please shorten (case report should be concise – see instructions for authors) : for instance, it seems irrelevant in context of equine hoof lesions whether in human medicine finger amputation is superior or not to local tumor excision (fingernails). However, the comparison with melanoma of the nail bed in man is important.

Lines 213-215 The authors cannot say with certainty that the first lesion was a keratoma without presenting histologic and immunohistochemical images.
It is reported that "the diagnosis of a keratoma should be based on the presence of horn growth in and around the hoof wall or sole and the characteristic microscopic findings of the lesion. Histopathologic examination of masses in the hoof region is essential for definitive diagnosis."
The conclusion is overstated. The authors fail to demonstrate unequivocally that "malignant transformation ranging from keratoma to melanocytoma to anaplastic melanoma occurred" without adequate histologic and immunohistochemical documentation. Figure 11 should be implemented with appropriate histology from samples obtained during all 3 surgical excisions; also, the magnification seems low and may not correspond to 40x magnification as indicated in the figure legend.  In the Conclusion, suggest changing or deleting last sentence with comparison of the lesion in people; suggest closing and concluding your report with a recommendation to veterinary clinicians and avoid venturing into comparisons for which evidence is lacking (consider that most of the time only Conclusions are read…).

Author Response

Thank you very much for your (very) thorough analysis. Many comments were valuable and were applied. Please see the attachment.

Round 2

Reviewer 2 Report

please see attached Word file

Author Response

Due to the reviewer's right remark, I added histopathological microphotograps of the primary lesion. In addition, I have added information on what basis this diagnosis was made (lines 226-241 revised version). Unfortunately, I do not have an immunohistochemical test, at that moment there were no indications to perform it, but I hope that this added histological specimen is sufficient evidence for a lesion in the form of a keratoma.

In revised version the term "low histological malignancy" was used 3 times (line 23, 144, 249) - it exists there since the original version of the manuscript and has not been changed (neither removed nor added).

Round 3

Reviewer 2 Report

Thank you for complying with suggestions and revising accordingly; please accept the following last small textual adaptions.

Title: the addition of “…”diagnosis of’….is not adding anything, so delete it.

Line 14: ….we describe the repeated excision of a hoof lesion, which…..and evolved over time into an anaplastic….

Figure 5 legend: …..diagnosis of keratoma. (delete “the”)